# Breast Cancer Diagnosis Using Extended-Wavelength–Diffuse Reflectance Spectroscopy (EW-DRS)—Proof of Concept in Ex Vivo Breast Specimens Using Machine Learning

**DOI:** 10.3390/diagnostics13193076

**Published:** 2023-09-28

**Authors:** Nadia Chaudhry, John Albinsson, Magnus Cinthio, Stefan Kröll, Malin Malmsjö, Lisa Rydén, Rafi Sheikh, Nina Reistad, Sophia Zackrisson

**Affiliations:** 1Department of Translational Medicine, Diagnostic Radiology, Lund University, 205 02 Malmö, Sweden; sophia.zackrisson@med.lu.se; 2Department of Medical Imaging and Physiology, Skåne University Hospital, 214 28 Malmö, Sweden; 3Department of Clinical Sciences Lund, Ophthalmology, Skåne University Hospital, Lund University, 223 62 Lund, Sweden; john.albinsson@med.lu.se (J.A.); malin.malmsjo@med.lu.se (M.M.);; 4Department of Biomedical Engineering, Lund University, 221 00 Lund, Sweden; magnus.cinthio@bme.lth.se; 5Department of Physics, Lund University, 221 00 Lund, Sweden; stefan.kroll@fysik.lth.se (S.K.); nina.reistad@fysik.lth.se (N.R.); 6Department of Surgery, Skåne University Hospital, 205 02 Malmö, Sweden; 7Department of Clinical Sciences Lund, Surgery, Lund University, 221 85 Lund, Sweden

**Keywords:** breast cancer, diffuse reflectance spectroscopy, extended-wavelength–diffuse reflectance spectroscopy, linear discriminant analysis, machine learning, support vector machine

## Abstract

This study aims to investigate the feasibility of using diffuse reflectance spectroscopy (DRS) to distinguish malignant breast tissue from adjacent healthy tissue, and to evaluate if an extended-wavelength range (450–1550 nm) has an advantage over the standard wavelength range (450–900 nm). Multivariate statistics and machine learning algorithms, either linear discriminant analysis (LDA) or support vector machine (SVM) are used to distinguish the two tissue types in breast specimens (total or partial mastectomy) from 23 female patients with primary breast cancer. EW-DRS has a sensitivity of 94% and specificity of 91% as compared to a sensitivity of 40% and specificity of 71% using the standard wavelength range. The results suggest that DRS can discriminate between malignant and healthy breast tissue, with improved outcomes using an extended wavelength. It is also possible to construct a simple analytical model to improve the diagnostic performance of the DRS technique.

## 1. Introduction

Breast cancer is the most common form of cancer among women and the second most common cause of cancer death globally [1,2]. In Sweden, it accounts for roughly 30% of all cancer cases among women [3]. The diagnosis is obtained using a triple-assessment method, which includes clinical examinations, radiological investigations, and a core-needle biopsy that yields a histopathological result. Conventional radiological investigations include mammography, ultrasound (US), and magnetic resonance imaging (MRI). However, each modality has its own benefits and limitations. The use of ionising radiation in mammography, the high user-dependency in US, and the cost and use of intravenous contrast agents as part of MRI are only a few such examples. The core-needle biopsy, which is the final part of the triple assessment, is an invasive procedure with an estimated false-positive ratio of 1–2% [4]. The procedure is often associated with discomfort, and the rarest and most severe complications include arterial bleeding, infection, and pneumothorax.

The search for a non-invasive technique that can provide relevant breast tissue diagnostic information in real time and without the use of ionising radiation or intravenous contrast agents has opened the door for optical modalities such as photoacoustic imaging (PAI) and diffuse reflectance spectroscopy (DRS) [5,6,7]. These modalities use a light source in the visible and near-infrared wavelength region to illuminate a biological tissue of interest. DRS has a portable and relatively simple instrumentation setup in comparison to PAI, making it a suitable first-line instrument for optical studies. In DRS, the illuminated light interacts with tissue through absorption and scattering. The absorption spectra depend on the chemical composition of the tissue (the assortment of molecules). The scattering spectra depend on the cellular morphology (the size of the molecules). Thus, by measuring the intensity of the diffusely reflected light, the concentration of different endogenous chromophores such as haemoglobin, lipids, water, and collagen can be obtained [7,8].

Previous DRS studies have attempted to create an “optical biopsy” for breast tissue by correlating spectral results with histopathology findings [5,7,8,9,10,11,12,13]. However, the number of studies is limited in this regard. This may be due to breast tissue showing considerable intra- and intersubject variation. It is, for example, morphologically heterogeneous, and it also exhibits structural changes, with varying degrees of lipid content in the various reproductive aging stages. This makes breast tissue a complex biological tissue [7,11]. In addition, many of these studies are based on DRS-spectra obtained in the “standard” visible to near-infrared wavelength range (VIS-NIR; ~450–900 nm) where haemoglobin and deoxyhaemoglobin are the major absorbers [8,13]. There are a few studies that have used an extended-wavelength (EW) range, including not only the VIS-NIR region but also the near-infrared and short-wave infrared range (NIR-SWIR, i.e., ~750–1600 nm). The added advantage is that the absorption peaks of water, collagen, and lipids are also included [12,14].

The majority of the breast-specific DRS studies use mathematical models, such as diffusion theory or Monte Carlo simulations, for data processing [5,7,8,9,10,11,12,13]. In contrast, there are DRS studies of other human organs, such as the cervix and liver, that have used multivariate statistical algorithms for data processing [15,16,17]. The overall advantage of the latter approach is that no prior knowledge of the absorption and scattering properties is required. To the best of our knowledge, there are no previous breast-related DRS studies that have used multivariate statistical algorithms for data processing.

In this work, we use a novel in-house-developed DRS setup that combines two types of spectrometers (VIS-NIR and NIR-SWIR) to visualise the EW range (~450–1550 nm). The combination of these two spectrometers covers most of the important chromophores in breast tissue in a single reading. This setup has successfully been used with liver and skin malignancies [16,18,19]. In the liver cohort, the main distinguishing feature between malignant and adjacent healthy tissue was observed in the visible-wavelength range [16].

This study aims to investigate the feasibility of using diffuse reflectance spectroscopy (DRS) to distinguish malignant breast tissue from adjacent healthy tissue, and to evaluate if an extended-wavelength range (450–1550 nm) has an advantage over the standard wavelength range (450–900 nm).

## 2. Materials and Methods

### 2.1. Patient Recruitment

Ethical approval was granted by the Swedish Ethical Review Authority (dnr 2019-04840) for an ex vivo experimental study conducted on breast specimens (total or partial mastectomy) taken from women undergoing surgery for primary breast cancer. This study was performed in accordance with the Declaration of Helsinki [20]. In total, 23 female patients who were scheduled for surgery at Skåne University Hospital, in Malmö, Sweden, were enrolled. Data collection was performed at the Department of Pathology and at Unilabs Breast Centre in December 2020 and May 2021. The optical measurements did not alter the standard clinical workflow. This study included women above the age of 18 with biopsy-verified breast cancer and a pre-operative mammography image showing a malignant breast lesion measuring at least 1 cm. Exclusion criteria included previous history of breast surgery or neoadjuvant treatment. Patients who did not comprehend Swedish were also excluded. Informed consent was obtained from all subjects involved in this study.

### 2.2. Instrumentation

A tungsten–halogen light source (Ocean Optics HL-2000-HP; Ocean Optics, Orlando, FL, USA) delivered a broadband spectrum (about 360–2000 nm) through a custom-designed fibre probe (10 mm in diameter) connected to a custom-made probe holder (25 mm diameter), as seen in Figure 1a. The fibre bundle had a central illuminating fibre (diameter 400 μm) encircled (diameter 5 mm) by ten collecting fibres (diameter 200 μm). Every alternate collecting fibre was attached to a spectrometer operating in the wavelength range of 350–1100 nm (Ocean Optics QE6500-VIS-NIR), and the remaining fibres were attached to a spectrometer operating in the wavelength range of 900–1700 nm (Ocean Optics NIRQuest512). All optical fibres had a numerical aperture of 0.22. The spectrometers’ slits of 50 and 25 μm, respectively, provided optical resolutions of about 3 nm. By using two spectrometers together, spectra were obtained in the range of 450–1550 nm (see Figure 1b) [16]. Both spectrometers enable real-time continuous spectra within their respective detector wavelength ranges. Computer software (Ocean View 2.0, Ocean Insight, Orlando, FL, USA) was used to operate the spectrometers and collect data on a laptop computer.

### 2.3. Data Collection

All measurements were performed on freshly excised breast specimens within 30–60 min of surgical resection. Both total and partial mastectomy specimens were used in this study. The partial mastectomy specimens were inked with different surgical dyes by the breast surgeon, as per clinical routine, with each colour representing a certain anatomical plane. Unfortunately, these surgical dyes limit optical measurements due to their scattering and absorption properties [21]. Thus, in the first two patients, DRS data were collected from total mastectomy specimens (non-skin-covered areas). In the remaining patients, the specimen was cut into ~5 mm thick slices by the pathologist, as per clinical routine, leaving the dye at the borders only, thereby allowing the use of partial mastectomy specimens as well. The slice in which the malignant tumour had its largest diameter was selected for EW-DRS measurements (see Figure 2a).

To minimise the effect of warm-up drift, the room temperature was recorded, and background and calibration spectra were recorded before the first measurement and after the last measurement, per specimen. The DRS probe was carefully positioned in direct contact with the malignant tumour, and if the tumour’s size was larger than the DRS probe, the probe was placed at multiple regions chosen at random. This was followed by measurements on adjacent healthy tissue chosen at random. On average, five separate locations were used to make DRS measurements in each tissue type. At each measurement site, a total of five optical measurements were obtained. The standard specimen mammography image was used as a reference to locate malignant tumour positions and adjacent healthy tissue, as seen in Figure 2b. Data collection for one measurement took about 12 s, and the total measurement time was set at about 20 min per specimen.

Patient demographic data (age and body mass index), core-needle biopsy results, the surgical technique used, and histopathological post-operative results were obtained from medical records. A protocol was set up to extract information from the standard pre-operative mammography (Figure 2c) and ultrasound report. Breast density was estimated based on the pre-operative mammography scan by two radiologists and classified according to the fifth edition of the Breast Imaging Reporting and Data System (BI-RADS) [22]. Variables such as tumour appearance and size were also noted.

### 2.4. Histopathological Analysis

Tumour characteristics were retrieved from pathology reports. Breast tumours were divided into five subgroups according to the 2019 WHO breast cancer classification system [23].

### 2.5. Multivariate Statistics and Machine Learning Discrimination Models

The included patient, radiological, and pathological variables are reported in numbers and percentages. Multivariate statistics and machine learning modelling were performed according to a previously described method used on liver malignancies [16]. More specifically, the spectral data were evaluated for all patients collectively and classified as either malignant or healthy tissue based on three steps. Firstly, a principal component analysis was performed to reduce the dimensions of data. Secondly, by using the first two principal components, the data were classified into either malignant or healthy tissue by using the linear diagnostic analysis (LDA) or the support vector machine (SVM) algorithm. The LDA algorithm increases the covariance between the two groups and decreases the variances within the groups. The SVM algorithm finds the hyperplane that can distinguish the data into two groups. Lastly, a cross-validation was performed using the “leave-one-out” method. The sensitivity (SE), specificity (SP), classification ratio (CR), and Matthew’s correlation coefficient (MCC) were calculated. Receiver operating characteristic (ROC) curves were plotted, and the area under the curve (AUC) was calculated. Data processing was performed in the statistics and machine learning toolbox™ package in MATLAB R2022a (The MathWorks, Inc., Natick, MA, USA).

## 3. Results

### 3.1. Demographics

The median patient age and the mean patient age were 66 and 67 years, respectively (range: 52 to 84 years). The median and mean BMI values were 28.7 and 27.2 kg/m^2^, respectively. About 8% of the women were using hormone-replacement therapy at the time of diagnosis. Invasive ductal carcinoma was the most common histopathological diagnosis, accounting for eleven patients (46%), followed by invasive lobular carcinoma, with eight patients (33%); tubular carcinoma, with two patients (8%); papillary carcinoma, with one patient (4%); and ductal carcinoma in situ, with one patient (4%) (see Table 1).

#### Pre-Operative Radiological Report

The average size of the breast malignancies was about 18.3 mm on both ultrasound and mammography (see Table 2). All breast density categories were represented with the most common breast density being B. For multifocal malignancies, the largest tumour diameter was used in the size calculations.

### 3.2. DRS Data

In total, 1035 EW-DRS spectra were obtained from 23 female patients: 505 from malignant tissue and 530 from adjacent healthy tissue (see Table 3). To enhance visual comparison, the means and standard deviations (±1 SD) of DRS for each tissue type were plotted against wavelength (see Figure 3). An overlapping wavelength region (930 to 1030 nm) of the two spectrometers was used to merge the two spectra into one continuous spectrum ranging from 450 to 1550 nm.

### 3.3. Machine Learning Outcome

Based on the wavelength range of 450–900 nm, the first two principal components, together, accounted for 83% of the total explained variance in the optical spectrum. When utilising the extended-wavelength range of 450–1550 nm, the corresponding figure was 80%. Whatever wavelength range used, almost all the spectral variation in the entire dataset was captured by the first two principal components. The linear discrimination analysis (LDA) and support vector machine (SVM) discrimination algorithms were used to categorise the samples based on only two principal components (see Figure 4). The score charts for the two first principal components show a clear discrimination between healthy and malignant breast tissue in the extended-wavelength range.

## 4. Discussion

The first objective of this ex vivo study was to evaluate whether EW-DRS can be used to distinguish between healthy and malignant breast tissue in an ex vivo setting. The results suggest that both tissue types have distinctive optical signatures. Malignant breast tissue displays increased absorption at a wavelength range of 950–1100 nm and decreased absorption at 680–950 nm and 1250–1550 nm, relative to adjacent healthy tissue. These wavelength ranges represent the absorption of essential breast tissue chromophores, such as haemoglobin (around 600 nm), water, lipids, and collagen (wavelengths around 1200–1400 nm) [14,24]. Thus, the EW range plays a vital role in characterising malignant and healthy breast tissue, and our results are consistent with previous DRS studies [5,10,13]. The multivariate approach does not, however, quantify the concentration of endogenous chromophores, making it an interesting topic for future studies.

An optical difference between malignant and healthy breast tissue is found in the NIR/SWIR wavelength range, as mentioned above. In contrast, malignant and healthy liver tissue demonstrates an optical difference in the VIS wavelength range (i.e., morphological differences are seen with the naked eye) [16]. This, in turn, can explain why breast surgeons face challenges when attempting to visually recognise tumour tissue at the surgical margin. It also highlights the crucial role of specimen mammography in detecting surgical margins.

Despite inter-patient variability, such as varying breast densities and different types of histopathological malignancies, a difference in the spectral signatures between malignant and healthy breast tissue is obtained. EW-DRS can detect a breast malignancy in mammographically dense breast tissue, which is a known radiological challenge. EW-DRS can detect common breast cancer types, such as invasive ductal carcinomas, as well as rarer forms, such as lobular and papillary carcinomas, with the former being a radiological challenge due to its mammographic features being similar to those of glandular tissue. Thus, DRS could play a complementary role in defining tumour borders and foci. This should be addressed in future studies with larger sample sizes.

The second objective of this study was to evaluate the role of an extended-wavelength (450–1550 nm), as opposed to the “standard” VIS-NIR optical wavelength range (450–900 nm), using a multivariate statical algorithm for data processing. The use of EW-DRS increases sensitivity from 33 to 92%, specificity from 70 to 90% and MCC from 3 to 82% using LDA, as well as from 40 to 94%, 71 to 91%, and 11 to 85% when using SVM, respectively. Similar results are observed for the receiver operating curves, as the AUC value is increased from 0.55 to 0.97 using LDA and from 0.57 to 0.97 using SVM when the extended-wavelength range is included. Previous studies of malignant and healthy breast tissue have reported sensitivities and specificities of about 90% and 88%, respectively [7]. The slightly improved results could be explained by the use of a multivariate and machine learning approach to data processing, as opposed to complex mathematical models. By using the extended-wavelength range and only two principal components, we managed to obtain relevant information, with a total of 80% variance being represented. This simplification is important for future clinical applications that require real-time processing.

Ex vivo studies like ours have limitations because tissue perfusion is eliminated after surgical resection. It has, however, been suggested that the difference in spectral signatures is not significantly different in ex vivo versus in vivo settings [9,10]. In other words, our ex vivo results may be applicable to future studies in which the breast is investigated pre-operatively (in vivo) by using other optical modalities with a greater depth penetration than DRS (e.g., PAI). Another potential limitation was that it was not possible to precisely correlate the final histopathological tumour borders with the placement of the DRS probe. However, guidance using the specimen mammography, as well as with the macroscopic contrast between the two tissue types being apparent, should have minimised incorrect measurements. Furthermore, our limited study sample included peri- and postmenopausal patients. However, all breast-density patterns were represented, and we have no reason to believe that the results would be different in pre-menopausal patients, because optical methods, in general, have shown that malignancy detection remains constant irrespective of breast density [25,26]. It should be noted that benign tumours were not included. The main reason for this choice was that these tumours are placed in a formaldehyde solution for fixation in the operating theatre as per clinical routine. This, in turn, alters their chemical and physical properties, making them unsuitable for inclusion in this study. Finally, the EW-DRS used in our study had a depth penetration of a few millimetres, depending on the wavelength [27]. This limits in vivo use during breast surgery, as the tumour itself is kept intact, surrounded by healthy tissue, upon resection. This DRS technique may be a more suitable real-time tool for pathologists to use in defining tumour borders or foci directly on tissue slices.

The results of this study will serve as an optical reference bank for future optical studies of breast tissue. It proves that EW-DRS can differentiate between malignant and healthy breast tissue.

## 5. Conclusions

We have shown that it is possible to distinguish malignant from healthy breast tissue using EW-DRS. Our results further suggest that it is possible to construct simple algorithms using only two principal components and standard machine learning discrimination algorithms such as LDA and SVM, thereby improving the diagnostic performance of the DRS technique.

## Figures and Tables

**Figure 1 diagnostics-13-03076-f001:**
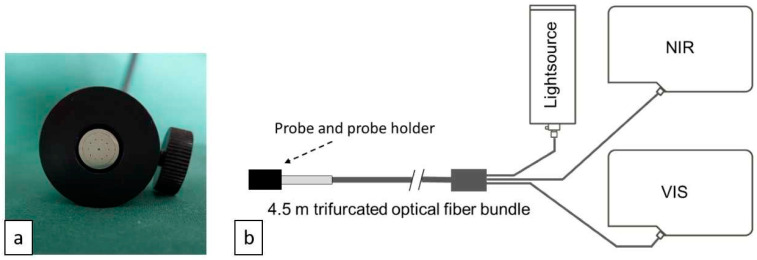
DRS instrumentation. (**a**) DRS probe (silver) attached to a probe holder (black). (**b**) The common leg consists of a central optical fibre connected to a light source and spectrometers. NIR, near-infrared spectrum; VIS, visible spectrum.

**Figure 2 diagnostics-13-03076-f002:**
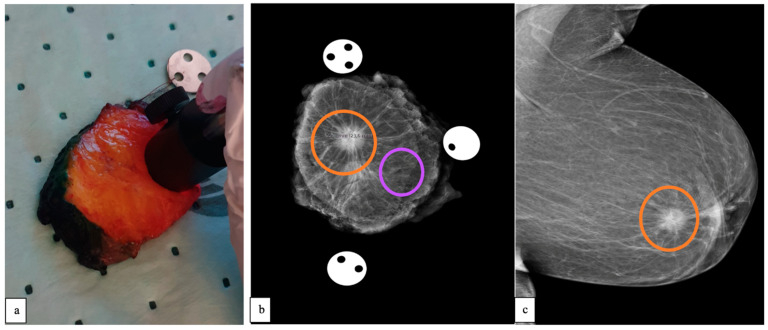
(**a**) Photograph showing the DRS probe during measurements on a ~5 mm thick partial mastectomy slice. (**b**) Mammography image of a partial mastectomy specimen showing a malignant tumour (orange circle) and adjacent healthy tissue (purple circle). (**c**) Pre-operative mammography in mediolateral oblique view showing the malignant lesion, which is marked with an orange circle.

**Figure 3 diagnostics-13-03076-f003:**
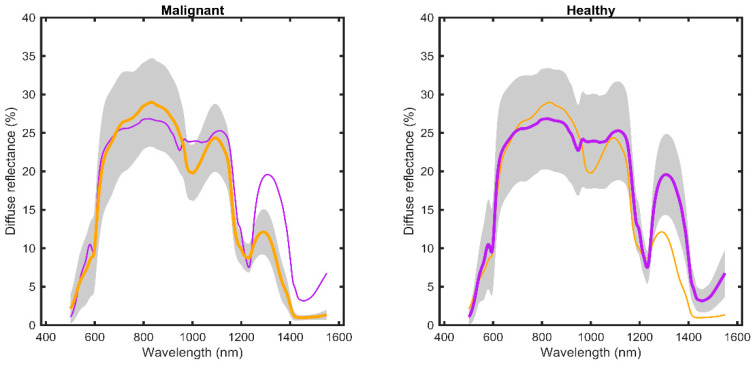
Diffuse reflectance spectra for all measurements from malignant (orange) and adjacent healthy (purple) breast tissue. Solid lines depict the mean intensities, and grey shaded areas represent ± 1 SD. A thin line showing healthy (purple) and malignant (orange) breast tissue has been added into the respective diagrams for comparison purposes.

**Figure 4 diagnostics-13-03076-f004:**
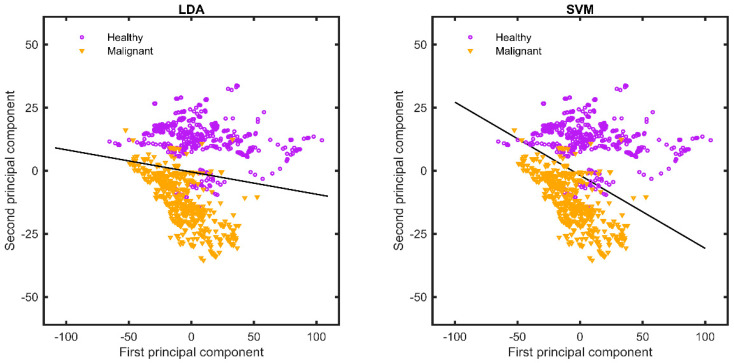
Score plots of two diagnostic parameters (first and second principal components) based on LDA (left) and SVM (right), respectively. LDA, linear discriminant analysis; SVM, support vector machine. The sensitivity (SE), specificity (SP), classification rate (CR), and Matthew’s correlation coefficient (MCC) values obtained using the LDA and SVM algorithms are listed in Table 4. A direct comparison can be made between the standard wavelength range of 450–900 nm and the extended-wavelength range of 450–1550 nm, with the latter showing improved values. The ROC curves and their respective AUCs are shown in Figure 5.

**Figure 5 diagnostics-13-03076-f005:**
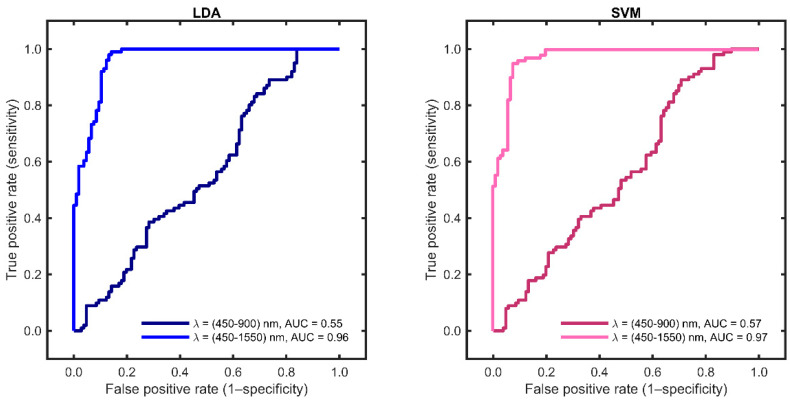
Receiver operating characteristic (ROC) obtained using LDA and SVM for the two wavelength (λ) regions, 450–900 nm (dark blue and dark pink) and 450–1550 nm (blue and pink). LDA, linear discriminant analysis; SVM, support vector machine; AUC, area under the curve.

**Table 1 diagnostics-13-03076-t001:** The demographics, pre-operative radiological imaging features, and histopathological results for each subject.

Patient nr	Age	BMI	HRT	Breast Density, BI-RADS 5th Edi	Mammogram (MAM) Tumour Appearance, mm	MAM Size, mm	US Tumour Appearance, mm	US Size, mm	Breast Specimen	Histopathological Diagnosis
1	68	20.3	No	B	Spiculated	10	Ill-defined, diffuse, hypoechoic	10	M	ILC
2	84	30.5	No	D	Partly ill-defined	16	Ill-defined, diffuse, hypoechoic	13	M	IDC
3	70	29.8	Yes	A	Indistinct, lobulated elongated	25	Hypoechoic	25	PM	IPC
4	54	30.8	No	A	Spiculated	11	Spiculated	10	PM	IDC
5	56	29.4	No	B	Ill-defined, diffuse	15	Ill-defined, diffuse, hypoechoic	15	PM	IDC
6	66	28.7	Yes	B	Ill-defined, diffuse	15	Ill-defined, diffuse, hypoechoic	11	PM	IDC
7	52	27.9	No	C	Spiculated, multifocal	15 + 10	Multifocal, ill-defined diffuse, hypoechoic	20	PM	IDC
8	71	34.6	No	A	Spiculated	17	Spiculated, hypoechoic	15	PM	ILC
9	77	18.5	No	D	Ill-defined, diffuse	*	Ill-defined, diffuse, hypoechoic	30	PM	IDC
10	84	29.5	No	A	Spiculated	18	Hypoechoic	14	PM	ILC
11	57	34.3	No	C	Multifocal	45	Multifocal, ill-defined diffuse, hypoechoic	36	M	ILC
12	52	29.8	No	B	Spiculated	18	Ill-defined, diffuse, hypoechoic	15	PM	IDC
13	69	26.6	No	A	Ill-defined, diffuse	10	Ill-defined, diffuse, hypoechoic	10	M	TC
14	71	29.0	No	B	Ill-defined, diffuse	10	Ill-defined, diffuse, hypoechoic	8	PM	TC
15	73	25.1	No	C	Partly ill-defined	40	Ill-defined, diffuse, hypoechoic	40	PM	ILC
16	57	18.3	No	D	Partly ill-defined	12	Ill-defined, diffuse, hypoechoic	12	PM	IDC
17	56	32.0	No	A	Calcification	20	Normal	*	PM	DCIS
18	61	26.0	No	C	Spiculated	12	Spiculated, hypoechoic	12	PM	ILC
19	72	33.5	No	C	Distortion	50	Ill-defined, diffuse, hypoechoic	60	M	ILC
20	70	23.5	No	B	Spiculated	12	Ill-defined, diffuse, hypoechoic	12	PM	IDC
21	56	20.5	No	D	Distortion	12	Ill-defined, diffuse, hypoechoic	8	PM	IDC
22	61	21.0	No	C	Partly ill-defined	17	Ill-defined, diffuse, hypoechoic	17	PM	IDC
23	74	26.0	No	B	Distortion	10	Ill-defined, diffuse, hypoechoic	10	PM	ILC

* Difficult-to-define borders, not measurable. BMI, body mass index; HRT, hormone replacement therapy; MAM, mammogram; US, ultrasound; OP, operation; M, mastectomy; PM, partial mastectomy; ILC, invasive lobular carcinoma; IDC, invasive ductal carcinoma; IPC, invasive papillary carcinoma; TC, tubular carcinoma; DCIS, ductal carcinoma in situ.

**Table 2 diagnostics-13-03076-t002:** Summary of the pre-operative radiological imaging features of all participants (*N* = 23).

Breast Density, BI-RADS 5th Edi (*n*, %)	
A	6 (26.1)
B	7 (30.4)
C	6 (26.1)
D	4 (17.4)
Ultrasound tumour size (mm)	
Minimum	10
Maximum	60
Mean *	18.3
Mammography tumour size (mm)	
Minimum	10
Maximum	50
Mean *	18.6

* The mean is based on 22 participants due to one tumour being non-measurable in respective imaging modality.

**Table 3 diagnostics-13-03076-t003:** Number of measurement sites and generated optical measurements.

Tissue	Number of Measurement Sites (*n* = 207)	Number of Optical Measurements (*n* = 1035)
Malignant	101	505
Healthy	106	530

**Table 4 diagnostics-13-03076-t004:** Summary of the machine learning outcome. The LDA and SVM diagnostic algorithms are represented at two wavelength ranges, the standard range between 450 and 900 nm and the extended-wavelength range between 450 and 1550 nm.

Diagnostic Algorithm	Wavelength Ranges	SE	SP	CR	MCC
	Nm	(%)			
LDA	450–900	33	70	52	3
	450–1550	92	90	91	82
SVM	450–900	40	71	56	11
	450–1550	94	91	92	85

LDA, linear discriminant analysis; SVM, support vector machine; SE, sensitivity; SP, specificity; CR, classification rate; MCC, Matthew’s correlation coefficient.

## Data Availability

The data supporting this study’s findings are available from the corresponding author upon reasonable request.

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
