# Peer review of "Breast Cancer Diagnosis Using Extended-Wavelength–Diffuse Reflectance Spectroscopy (EW-DRS)—Proof of Concept in Ex Vivo Breast Specimens Using Machine Learning"

_diagnostics, 2023, doi:10.3390/diagnostics13193076_

Round 1
Reviewer 1 Report
In this manuscript, the authors proposed using diffuse reflectance spectroscopy (DRS) to distinguish malignant breast tissue from adjacent healthy tissue and have investigated the suitable wavelength range. The diagnosis of tumour tissue is much important during the surgery. The quickest way to identify the malignant breast tissue provides the most potential to reserve the quality of survival of patients. The manuscript still has some issues need to be revised and addressed before accepted, as I listed below.
1. Can you provide the spectral curve of the light source? The illumination intensity is different at every wavelength, will it has influence on the spectra detection? Have you tried to calibrated it?
2. What does the “Every second collecting fibre” means in the Section 2.2?
3. Is the response sensitivity of the two spectrometers the same? What kinds of detectors they use?
4. During the measurement, is the light has an influence on the tissue changing? As the tungsten-halogen light source always causes the temperature increasing during the detection process. What is the distance between the probe and the sample surface?
5. The division of the training set and test set needs to be further explained in detail. Is it divided by 1035 spectrals after mixing? Or the spectra from some patients are used for training and others for testing?
6. In Figure 3, the spectrum obtained by the two spectrometers should be given separately.
7. Some superscript or subscript formatting needs adjustment.
Author Response
We are truly grateful for the thorough reading and rigorous review by the two reviewers. We have revised the manuscript in accordance with our understanding of their feedback, suggestions, and good advice.
- Can you provide the spectral curve of the light source? The illumination intensity is different at every wavelength, will it have influence on the spectra detection? Have you tried to calibrate it?
Thank you for this important feedback. The spectral output of the light-source can be found on Ocean Insight's webpage at this link: https://www.oceaninsight.com/globalassets/catalog-blocks-and-images/catalog-blocks-and-images/light-sources/visible-and-nir-light-sources/hl-2000-tungsten-halogen-sources-spectral-output.pdf
The calibration was conducted using the standard method described in: Reistad, N., & Sturesson, C. (2022). Distinguishing tumor from healthy tissue in human liver ex vivo using machine learning and multivariate analysis of diffuse reflectance spectra. Journal of Biophotonics, 15(10), e202200140. The calibration information has been added on page 4, row 138-140.
- What does the “Every second collecting fibre” means in the Section 2.2?
We have changed it in line with your input, thank you. There are 10 collecting fibers in the probe, with alternate fibers connected to each spectrometer. This information has been clarified on page 3, row 107-110.
- Is the response sensitivity of the two spectrometers the same? What kinds of detectors they use?
Thank you for great questions and showed interest. No, it is two different types of detectors. The QE6500 Pro spectrometer is equipped with a Hamamatsu FFT-CCD detector that has 1044 pixels and is responsive from 200-1100 nm. The NIRQuest512 spectrometer features a Hamamatsu high-performance InGaAs array detector with 512 pixels, and it has a detector range of 850-1700 nm. Detailed information about the detectors can be found at the producers’ webpage.
- During the measurement, is the light has an influence on the tissue changing? As the tungsten-halogen light source always causes the temperature increasing during the detection process. What is the distance between the probe and the sample surface?
We have clarified it in line with your input, thank you. The very slight temperature increase during the measurement has no significant impact on the spectral signature.
The probe is in direct contact with the tissue throughout the measurement. This information has been further clarified on page 4, row 140-141.
- The division of the training set and test set needs to be further explained in detail. Is it divided by 1035 spectral after mixing? Or the spectra from some patients are used for training and others for testing?
Thank you for your feedback. Cross-validation was carried out using the “leave-one-out” method. This information has been clarified on page 5, row 170.
Details can be found in: Reistad, N., & Sturesson, C. (2022). Distinguishing tumor from healthy tissue in human liver ex vivo using machine learning and multivariate analysis of diffuse reflectance spectra. Journal of Biophotonics, 15(10), e202200140.
- In Figure 3, the spectrum obtained by the two spectrometers should be given separately.
Thank you for your feedback. We believe that providing the two spectra individually does not add any value to the analysis. We have, however, clarified that we used the overlapping wavelength region (930 to 1030 nm) of the two spectrometers to merge the two spectra into one continuous spectrum ranging from 450 to 1550 nm. This information has been added on page 7, row 205-207.
- Some superscript or subscript formatting needs adjustment.
Thank you for noting. The superscript or subscript formatting has been adjusted according to the reviewer’s comment.
-----------------------------------------------------------
Reviewer 2 Report
Chaudhry et. al. have presented the application of diffuse reflectance spectroscopy in identifying malignant tumor in breast tissue. The novelty claimed in the present work in use of extended wavelength region and analysing data using machine learning algorithms. The results are interesting and the approach offers a non-invasive diagnosis for breast cancer, though this isn't demonstrated in the present work.
Some points to address for improving overall readability.
1. The introduction should explain more on diffuse reflectance spectroscopy.
2. In line 109 optical resolution of 3 nm is discussed. What is the spatial resolution of this approach? Do the authors scanned in steps of 3 nm in range 450-1600 nm or was it something else?
3. We can see remarkable difference in 1200-1400 nm range to distinguish between malignant and healthy tissue. Can authors use the ML algorithms on the data from 1200-1400 nm and see if there is any improvements? It will be good to know which chromophores are responsive in that range.
4. Please describe the feature matrix (in detail) used in the PCA and subsequently in SVM or LDA.
Author Response
We are truly grateful for the thorough reading and rigorous review by the two reviewers. We have revised the manuscript in accordance with our understanding of their feedback, suggestions, and good advice.
- The introduction should explain more on diffuse reflectance spectroscopy.
Thank you for your valuable feedback. This information has been added on page 2, row 50-55.
- In line 109 optical resolution of 3 nm is discussed. What is the spatial resolution of this approach? Do the authors scan in steps of 3 nm in range 450-1600 nm or was it something else?
Thank you for the feedback and valuable input. The spatial resolution was not explored in this particular study and would be an interesting topic for a future study.
The QE6500 Pro spectrometer is equipped with a Hamamatsu FFT-CCD detector that has 1044 pixels and is responsive from 200-1100 nm. The NIRQuest512 spectrometer features a Hamamatsu high-performance InGaAs array detector with 512 pixels, and it has a detector range of 850-1700 nm. Both detectors can be cooled using onboard TE-Coolers to minimize dark noise. Additionally, both spectrometers enable real-time continuous spectra within their respective detector wavelength ranges, thus no scanning steps are applied. This scanning procedure has been clarified in page 3, row 113-115.
- We can see remarkable difference in 1200-1400 nm range to distinguish between malignant and healthy tissue. Can authors use the ML algorithms on the data from 1200-1400 nm and see if there are any improvements? It will be good to know which chromophores are responsive in that range.
Thank you for your feedback. In the 1200-1400 nm range, the main absorbing chromophores are water, lipid, and collagen. There are however also differences in other wavelength ranges that might be of interest. This is a matter for a more detailed analysis in a future study and hence out of the scope of the present study.
This information has been added on page 9, row 248-251.
- Please describe the feature matrix (in detail) used in the PCA and subsequently in SVM or LDA.
Thank you for you feedback. We utilize features extraction by PCA as input for a classification algorithm, either based on linear discriminant analysis (LDA) or a support vector machine algorithm (SVM). We are uncertain about whether your question is applicable in our analysis. An attempted clarification has been added to page 4-5, row 164-172. Did we understand your question correctly? If not, could you please give us some more information?